# Discussing Human and Environmental Health Co-Benefits Related to Diet and Mobility Behaviours in the Primary Care Setting: A Qualitative Exploratory Study

**DOI:** 10.3390/ijerph22101503

**Published:** 2025-09-30

**Authors:** Aline Sigrist, Elodie Richardet, Nicolas Senn, Joëlle Schwarz

**Affiliations:** Center for Primary Care and Public Health, Unisanté & University of Lausanne, Pré-du-Marché 23, 1004 Lausanne, Switzerland; elodie.richardet@gmail.com (E.R.); nicolas.senn@unisante.ch (N.S.); joelle.schwarz@unisante.ch (J.S.)

**Keywords:** health and environment co-benefit, climate change, general practitioners’ role, medical encounter, primary care, patients’ acceptability

## Abstract

The interconnection between health and environment is increasingly recognised, as is the role of healthcare professionals in raising awareness among patients and healthcare policymakers. To explore the relevance of enhancing patients’ awareness of the links between health and environment in ambulatory care, we conducted a qualitative study on General Practitioner (GP) practices in Switzerland. Using a co-benefit approach, we designed materials on meat consumption and active mobility, which were displayed in the waiting rooms. We conducted observations in five practices and interviewed five patients and five GPs to examine patients’ understanding of the messages and assess the acceptability of discussing them during medical encounters. Patients and GPs were receptive to the co-benefit approach. However, barriers were identified in promoting co-benefits during clinical encounters, including time constraints and lack of knowledge. Patients showed reactance to the messages and questioned the reliability of messages related to climate change. GPs were reluctant to compromise relationships with patients and were ambivalent, viewing environmental discussions as political. Positive message display and community promotion were identified as facilitators. This study highlights the need to develop clear educational materials to support GPs, to adapt messages to patients’ backgrounds, and to address the dichotomy between GPs’ political and scientific perspectives.

## 1. Introduction

The evidence of close links between global environmental degradation, including climate change, and health, is increasingly being recognised at the public health level [1,2,3,4,5], although there is a lack of public awareness of these links [6]. Climate change is widely regarded as the most significant threat to human health [7] and is expected to further impact clinical practice through the emergence of new diseases or the increased prevalence of climate-related pathologies [8]. In the health sector, as in society at large, two roles are portrayed related to climate change: adaptation, by anticipating and reducing the threats on human health due to climate change (i.e., public health preparedness), and mitigation, by reducing the carbon emissions of the health sector itself on one side, and on the other side by influencing human behaviour to reduce practices that contribute to climate change. The latter include actions to decrease the prevalence of diseases such as chronic diseases that require care with a high environmental impact [9].

Regarding this latter role, studies have highlighted the urgency for action and the important contribution that health professionals can make in disseminating information and raising awareness among populations and health policymakers [5,8,10,11,12,13,14,15]. The ability of health professionals to engage with individuals and promote preventive behavioural changes, such as healthy habits (e.g., smoking cessation) is viewed as a valuable skill that professionals can use in their clinical practice. It enables them to influence patients’ behaviours in a way that benefit both the environment and human health [8,12,16,17]. To support this approach, the concept of co-benefits is employed to emphasize the reciprocal advantages for human health and planetary well-being that result from strengthening the connections between environment and health. This concept is defined as the positive health outcomes that emerge from interventions aimed at mitigating environmental degradation or, conversely, from initiatives that promote environmental preservation while simultaneously enhancing human health [12]. Among others, Hopkins et al contributed to a better understanding of the concept of co-benefits, which they referred as “win-win interventions” in their publication [18]. Family medicine and primary care settings appear particularly well positioned to provide information and support action on the links between individual and ecosystem health. Indeed, it is a setting close to patients and the population where prevention and health promotion activities can be carried out.

Identifying and promoting the co-benefits of climate change strategies can provide a positive framework for such interventions by highlighting the opportunities arising from the societal changes needed for ecological transition [19]. In this sense, the World Organization of National Colleges, Academies and Academic Associations of General Practitioners/Family Physicians (WONCA) has identified diet and mobility as two topics that could be addressed during clinical encounters, with a view of benefiting both the health of the population and the environment [20]. Diet and personal mobility account for 20–40% and 30–40% of greenhouse gas emissions, respectively. According to the EAT-Lancet report, diet is a key lever for optimising both health and sustainability. To achieve a healthy diet by 2050, meat consumption must be reduced by 50% [19].

A systematic review has summarised the current evidence on barriers and facilitators for physicians to adopt a patient–planetary, co-benefit approach to health [21]. Barriers included the absence or scarcity of resources, including perceived insufficient knowledge and education of physicians on the topic, absence of clear and ubiquitous clinical recommendations and educational material, and limited time to engage in co-benefit prescribing, related to the inability to bill people for such activity. Facilitators included existing policy statements and guidelines from reputable institutions, as well as available education materials. A study included in the systematic review was conducted by André et al. in Switzerland [8]. It explored general practitioners (GPs)’ knowledge and attitudes on climate change and health, and their interest in promoting a co-benefit approach in their clinical practice with patients. The study demonstrated that GPs believe patients should be better informed about the links between global warming and health, that it could be part of the medical consultation, and that some were already discussing climate change with their patients. Corroborating these with other existing evidence, the perceived barriers were lack of time, of scientific recommendations or even of knowledge. The study also suggested that the political orientation of GPs played a role in their willingness to adopt a co-benefits approach.

Following up on the study from André et al. [8], and responding to the perceived need for educational material, a project was launched in September 2021 in the GP practices of Western Switzerland that aimed to develop material related to the co-benefits to be displayed and used in GP practices. By displaying the material in waiting rooms, the intervention objective was to trigger interest in patients, discussions between patients, and an opportunity for GPs to discuss the topic of meat consumption and active mobility with patients during the consultation. Going beyond André et al.’s study on knowledge and attitudes [8], this study assessed the implementation of a co-benefits approach in real-world practice. The aim of this study was to qualitatively assess the implementation of a co-benefits approach in real-world practice. We specifically aimed to assess (i) the understanding and perception of the co-benefits’ messages from the point of view of the patients and (ii) to examine GPs’ perceptions of their role and legitimacy in addressing co-benefits with patients during a clinical encounter.

## 2. Materials and Methods

### 2.1. Study Setting and Design

This qualitative study was conducted in family medicine practices in Western Switzerland between September 2021 and December 2022. To explore patients’ understanding and perception of the co-benefits messages displayed in the form of posters and factsheets in the waiting rooms of GP practices, we carried out observations in the waiting rooms, as well as informal individual or group discussions on these materials with patients awaiting their consultation. To further assess the acceptability of engaging in a co-benefits discussion during clinical encounters, we conducted follow-up, individual, semi-structured interviews with patients recruited during the observations and with GPs, as well as a focus group discussion (FGD) with GPs who participated in the study. As no personal data on health were collected, this study was exempt from ethics committee clearance (Req-2022-00160 from the Ethics Commission of Canton de Vaud).

### 2.2. Study Material

This study was conducted through a partnership approach involving a university institution and a non-governmental organisation named *Engagés pour la santé* [Committed to Health]. Accordingly, the research material was developed in close collaboration with the local healthcare association. The co-benefits material was developed by the project team (authors of this paper, Julia Gonzalez Holguera, and four health professionals from the Swiss association *Engagés pour la Santé*, namely two GPs, one physiotherapist, and one psychiatrist—all women). Collaborative working sessions were held to define the focused co-benefits topics and the optimal format for an intervention in a waiting room. The team decided to focus on two topics, meat consumption and active mobility, and to develop two types of support material: posters and a factsheet (the study material is available in the Appendix A). The two posters, one for each topic, were created using existing material developed by local healthcare professionals’ associations—*Engagés pour la Santé* and *Médecins en faveur de l’environnement* [Doctors for the environment]—involved in co-benefits approaches. During the working sessions, the team discussed the posters’ content and visuals, which were adapted and reviewed at the following working session, until they reached a consensus. The team modified the content to avoid guilt-inducing messaging and to ensure design consistency with the other project materials. The factsheet used the same existing material, supplemented with additional information from the EAT-Lancet Commission data [19].

### 2.3. Sampling Procedure

Family medicine practices from Western, francophone Switzerland were contacted by email and invited to take part in the study, presented as a follow-up of André et al.’s study [8]. Practices were selected from a list of the Department of Family Medicine (DFM) that groups GPs collaborating with the department for education, clinical supervision, or research activities. A total of 65 practices were contacted across four Cantons (Vaud, Neuchâtel, Jura and Genève). Eight practices responded, expressing interest in participating in the study. However, two practices eventually withdrew during the phase of the observation (as discussed in Section 3). The study was therefore conducted in six practices located in the Cantons of Vaud and Neuchâtel. Medical practices were thus recruited using a convenience sampling procedure (see Table 1 for a description of medical practices). Patients attending consultation days were included in the study. During the observations, all patients present in the waiting rooms were invited to share their reaction to the information material. In each medical office, a small number of patients (between 1 and 3) declined to participate in the discussion. Patients who expressed interest in further discussing the topic were recruited for pursual, with individual interviews to be conducted in their preferred location—their home or the DFM office. Many declined due to lack of interest or of time. To complement patient recruitment, one participating GP recruited and organised 3 patient interviews, which were conducted at their practice. Regarding GP interviews, all participating GPs were invited to take part in an individual interview, as well as in a final FGD to collect their experience and perceptions of the project. One GP was unavailable for the interview.

### 2.4. Data Collection

The observations and informal discussions with individual or groups of patients were conducted by AS and ER, both female health anthropologists, over half a day or a full day, depending on the availability of GPs and the patient flow. Patients present in the waiting room were prompted on their interest in the displayed posters and factsheet, on their understanding of and reaction to the messages and the relevance of finding such information in this location. Observations and informal discussions were transcribed into notes. The semi-structured interviews with patients aimed to delve deeper into the questions raised during the observations. They were conducted by AS or ER, using an interview guide developed by the project team that included items such as perception of the relevance of addressing climate issues in the clinical practice, perception of the displayed material, and legitimacy and role of GPs in addressing these topics during a clinical encounter. Interviews with GPs were conducted by AS or ER and aimed to explore their opinions about the study and project material, how they perceived their legitimacy and role in addressing climate issues, and co-benefit approaches with patients. All interviews were conducted in French, audio-recorded, and fully transcribed. The quotes used in this article were translated into English. Finally, the FGD with GPs conducted by AS sought to interpret the preliminary findings from the observations and interviews that were presented, as well as to gather propositions on how to integrate a co-benefits approach in clinical practice. Notes were taken by JS. Elements from this FGD are presented in Section 4, in the list of possible avenues to integrate a co-benefits perspective in family medicine practices.

Throughout the process of data collection, AS and ER presented and reiterated their position as anthropologists engaged in a project promoted by health professionals. Despite being involved in the development of the intervention material, they clearly did not position themselves either in favour or against the use of such material in family medicine practices during informal discussions and interviews, in order to ensure a neutral position and foster patient reactions.

### 2.5. Data Analysis

Data were analysed by AS and ER using thematic analysis [22]. Conducted as an iterative and continuous process, the initial analysis informed subsequent data collection and further analytical steps. The analytical process began with the observation notes, which constituted the initial dataset. AS and ER jointly analysed these notes, as they had conducted the observations together. They collaboratively identified salient features, coded relevant segments, and collated them into preliminary themes. These themes were then discussed and refined with the other two authors (NS, a male physician and epidemiologist, and JS, a female sociologist and epidemiologist). They were subsequently used to guide further analysis during the individual interviews. In a second phase, the interview transcripts were fully analysed separately by AS and ER. Coding was conducted using the previously identified themes, while allowing for the inductive emergence of new themes. The new and final themes were compared and discussed in conjunction with NS and JS. The final FGD was used to reach consensus on the final thematic map and ensure analytical coherence. Elements brought by FGD participants were used to develop the recommendations presented in this article. Finally, the entire dataset was comprehensively reviewed using the final thematic map, with a triangulation of findings from observation, semi-structured interviews, and FGD. The analytical process was regularly discussed within the team to maintain coherence and consistency.

## 3. Results

In total, eight family medicine practices accepted the invitation to participate, situated in two Cantons: Vaud and Neuchâtel. Two practices withdrew: one ultimately refused to display the poster on meat consumption because it was perceived as too controversial for their practice (this is discussed below), and one no longer replied to the researchers’ solicitations. Activities in the six practices were observed, all in urban or semi-urban settings (see Table 1 for a description of medical practices and participants’ profiles). An average of ten patients per practice were present during the observations. A total of 41 patients (27 women and 14 men) participated in informal discussions with the researchers. Five patients were followed up with a semi-structured interview (three women, two men), that lasted from 30 min to two hours. Five interviews were conducted with GPs (three women, two men), also lasting 30 min to two hours. Three women GPs participated in the final FGD. In the next sections, we present the findings from these different sources of data, developing three main themes: (1) patients’ reactions to the approach and to the specific topics; (2) the perceived role and legitimacy of GPs in addressing climate issues within their medical practice; and (3) the propositions formulated by participants to improve the approach.

### 3.1. Participants’ Reactions to the Approach and Messages

The observations conducted in the waiting rooms showed that most patients did not spontaneously pay attention to the information material displayed in the medical practices. The poster that focused on meat consumption drew more interest and discussions with the researchers than the one on active mobility. The reaction to the factsheet varied; some GPs had to ask for more factsheets to be sent for displaying purposes because they were taken home by patients, while in other waiting rooms there seemed to be little to no interest in them. Some patients commented on the lack of legibility of the factsheets, expressing that the information was too technical and complex, or abstract. We observed that the impact of the information material appeared to differ according to the environment of the waiting room. In practices where much health prevention documentation was displayed, the posters and factsheet tended to be hidden by other information, while practices with a more sober decoration style, with nature, or travel pictures made the information material more visible.

Informal discussions and follow-up interviews revealed that patients clearly identified the topic of environmental health in the posters and factsheet but less the topic of individual health and the co-benefits perspective intended. Thus, the reactions collected were rather related to the acceptability and pertinence of thematising climate change in a medical setting overall, and the link to their own health was not evident at first. It is interesting to note that in general, patients reported being sensitive to climate change and in favour of the approach proposed, but when discussing the matter more in-depth they tended to express that they preferred focusing on their own health during the consultation.

In the following sub-sections, we present three sub-themes that emerged from analysis of participants’ reactions to the approach, namely (i) the controversy of messages around meat consumption; (ii) the variability in reactions across social and cultural repertoires of participants; and (iii) the expressed perception that individual action and responsibility is limited with regard to the topic of environmental health.

#### 3.1.1. The Controversy of Meat: Is It Good or Bad?

We found that the topic of meat consumption was more controversial in comparison to active mobility. Most patients had the previous notion that fatty meat has negative impacts on health, mainly through its association with increased risk for cardiovascular disease and colon cancer. At the same time, however, participants also mentioned that there are negative health consequences associated with the absence of meat consumption, related to protein deficiency, especially with certain categories of persons. In fact, the GP practice that refused to take part in the study when discovering the poster and information on meat consumption justified their withdrawal by explaining that meat consumption was important in the diet of some of their patients, who needed regular protein intake. Furthermore, a general perception of patients was that some categories of persons such as older persons or people engaging in sport need protein to maintain their muscles, and thus recommendations to reduce meat consumption were not perceived as appropriate. Discussions on meat consumption also took place in relation to the topic of animal exploitation, whereby reducing consumption enabled reducing exploitation. Participants also raised the notion that the quality and origin of the meat was of importance, and this could be more nuanced in the information displayed. Some GPs reported ambivalent positions with regard to the link between meat consumption and climate change: 

“*I don’t know if you saw the article in yesterday’s paper, (...) there was a quote from an agronomist stating that eating vegan had no less effect on the climate than eating like omnivores.*”(GP05)

#### 3.1.2. Social and Cultural Repertoires

Our study revealed variability in the reception and perception of the information disseminated, based on sociocultural background, socioeconomic position, gender, health status, age, and household composition. Age appeared to be an important factor influencing the perception of meat consumption. Among older people, some were sensitive to the future of their grandchildren and thus adhered to the messages on the need to reduce meat consumption. The generation of young adults appeared sensitive to climate issues and reported being aware of the topic since their childhood. We found that in the age group of people around 40 years old, greater ambivalence was present. Some presented themselves as meat lovers and could conceive reducing their consumption only with great difficulty. Inversely, others expressed an eco-anxiety that influenced their everyday habits in the sense of abandoning meat consumption altogether. Similarly, beliefs that vehiculated around the benefits of protein consumption for certain categories were often related to people’s age and gender, such as growing children, “working men”, and older persons. This older patient expressed his concern about decreased meat consumption and anaemia, for example:

“*Before we ate the whole animal, we ate the offal. It was cheaper and good for health, especially for anaemia, especially calf’s liver for example. We tried to solve anaemia problems without eating meat, but it didn’t work.*”(PA)

Beyond age categories and gender, we also found that sociocultural repertoires of participants modulated their perceptions of eating habits. People who had experienced low meat consumption during the war and/or post-war period in which they grew up had the habit of not consuming meat on a regular basis, and thus reported that low meat consumption was easily achievable. Inversely, other participants expressed that, because they were restricted in accessing meat in childhood, they were not willing to give up meat now that it was accessible to them, especially in older age. Similarly, participants who originated from sociodemographic backgrounds where meat access was limited or where climate change is not thematized reported that changing their consumption habits was challenging. A patient originally from Ethiopia explained that in her country climate change is not a concern, as there is no public awareness on the issue. She explained that when living in Ethiopia she cooked more vegetables and little meat. Living now in Switzerland, she appreciated spending little time in cooking due to her demanding professional activities, and this induced cooking less vegetables and more meat. Another participant from Cameroun explained that in Switzerland she eats more meat because she did not have the opportunity in her country due to financial constraints. As meat was financially accessible to her in Switzerland, she was not willing to constrain herself and her family.

#### 3.1.3. Reactance to Messages Emphasising Individual Responsibility

After discussions with patients in the waiting rooms, we found that those who already appeared sensitive to and concerned about climate issues were in favour of displaying such information and messages in medical practices. In contrast, patients who seemed less engaged with the topic perceived the posters as too moralistic, placing undue emphasis on individual responsibility, and thereby inducing feelings of guilt and powerlessness. Paradoxically, the informational material triggered counter-reactions among those least sensitive to climate change issues—even though they were precisely the target audience for these messages, intended to encourage changes in attitudes and behaviours. Some participants reacted negatively to the displayed information, perceiving it as overly moralistic, polarised, and even anxiety-inducing:

“*It’s green propaganda, I’m sick of it! We don’t know what’s good and what’s not. They restrict freedom of action and freedom of expression—soon we won’t be able to do anything anymore.*”(PC)

Another dimension of reactance expressed by participants was the perceived emphasis on individual responsibility. Some patients reported that the displayed messages overemphasised personal responsibility, which they viewed as unfair given the significant role of industry. Many participants pointed out that industrial activity has a far greater environmental impact than individual actions, and they expressed a sense of unjust guilt triggered by messages focusing on personal behaviour:

“*It’s depressing. We know it. As citizens, we do our best, but it’s the industry that needs to be regulated.*”(PB)

“*We should be targeting fast food chains that use industrial meat or supermarket that use plastic packaging.*”(PE)

This view—that messages targeting individual behaviour are controversial in light of the industry’s burden—was also shared by some GPs:

“*It is an individualistic approach. I’ve been raising patient awareness for 20 or 30 years now, and it’s true, I see the progress, which is good! I’m not denying that, but it takes a lot of time, and I believe that if society doesn’t take more responsibility, we will never achieve it.*”(GP03)

In addition, patients expressed a sense of powerlessness in effecting change. Individual actions were seen as insignificant in the face of a neoliberal economic system that prioritises economic growth over effective and sustainable environmental measures. Patients highlighted a lack of overall coherence in climate-related efforts, particularly regarding the role of the economy, which contributed to a general feeling of discouragement:

“*We feel powerless because, in the end, the economy always has the last word.*”(PB)

“*I’m going to be defeatist, but it’s lost in advance.*”(PE)

Some GPs also emphasised the influence of vested interests, particularly within pharmaceutical lobbies.

Beyond the negative reactions to the content of the displayed messages, some participants found the approach meaningful and useful. Several expressed that the waiting room or the medical consultation was an appropriate setting for displaying information that could serve as reminders in a long-term process:

“*It’s almost like a socialization to a different way of eating. It doesn’t happen overnight.*”(P04)

“*We know, but we don’t do it. We are all full of good intentions, but daily life resumes and we need reminders.*”(PA)

### 3.2. Role and Legitimacy of GPs in Addressing Climate Issues in Clinical Encounters

The informational material displayed in the waiting room had a little impact on medical consultations. Only one GP reported that the project had enabled patients to engage in discussions about health and the environment. However, the presence of the two researchers in the waiting room appeared to encourage patients to raise these topics during consultations.

Interestingly, although most participating GPs expressed alignment with the importance of discussing co-benefits with patients, they were ambivalent about their legitimacy and capacity to do so in practice. Some patients also shared their views on the appropriateness and ability of GPs to address co-benefits during clinical encounters.

#### 3.2.1. “No Political Opinions in My Consultations!”

One reason for ambivalence that emerged during observations and interviews was the tension between the topic of the environment—perceived as political—and individual health—perceived as private and a priority. Some GPs expressed that discussing the environment in general during consultations was problematic:

“*I’m not the arm of health policy... I’m not very comfortable with anything political...*”(GP02)

“*We must not politicize health. Health is a right for everyone.*”(GP05)

Some GPs suggested that focusing solely on individual health benefits was sufficient, and that environmental benefits could remain unspoken:

“*I’m trying to tell [this patient] that it’s for his own health that these changes are beneficial. I don’t talk much about the methane produced by the cows and so on. I have always tried to refrain from giving political opinions in my consultations.*”(GP04)

As expressed in the last quote, some GPs were aware of co-benefits but preferred not to discuss environmental aspects due to their political tinge. One GP explained that talking about the environment is inherently political, and engaging on this topic inevitably involves political positioning:

“*We do have some data, but they are interpreted differently by political parties (...) doctors may approach them differently depending on their political affiliations.*”(GP04)

Other GPs, however, expressed that discussing the environment was acceptable if framed through patient- and health-related co-benefits:

“*The environment via health issues, yes—but the environment itself, no. I’m not the one to say the environment is a problem. I’m not the one to tell patients whether they should travel by plane or not.*”(GP02)

#### 3.2.2. Preserving the Therapeutic Alliance

One reason the GPs gave for their reluctance to discuss environmental benefits—even from a co-benefits perspective—was rooted in professionalism and concern about jeopardising their therapeutic alliance with patients. Building and maintaining a trusting relationship is a cornerstone of medical practice, and statements on environmental topics that could be perceived as judgmental were seen as potentially harmful to that trust:

“*For me, it must not be political. Because we have to maintain a certain credibility with all our patients. And it’s important that we don’t judge the person who comes with his polluting 4-wheel drive.*”(GP05)

This concern was shared by some patients, while others expressed trust in their GPs to provide valuable and reliable information on environmental health, as illustrated in the following quotes:

“*It depends on how you approach it. It’s not the doctor’s place to impose their views. When people come to a doctor for health problems, they shouldn’t feel that the doctor is trying to convince them of their point of view.*”(P)

“*That’s what’s important for me with my family doctor... He’s not just someone who gives a prescription or medicine when you have a fever. He’s someone who gives good advice, who is knowledgeable, who accompanies me. They have a lot of information, and it’s their job to have it—so I think their advice is important...*”(P05)

These two perspectives illustrate that ambivalence and differing opinions also exist among patients regarding the role and legitimacy of GPs in addressing climate issues.

#### 3.2.3. Who Pays for Talking About Co-Benefits?

Finally, another reason for reluctance in discussing environmental benefits alongside individual health benefits was related to time and cost, as expressed by both GPs and patients. Participants raised concerns that adding an environmental dimension to consultations would require additional time, which—under the current health financing scheme—would need to be covered by patients. As a result, some patients responded that they were not willing to pay more for climate advice from their GPs:

“*If it’s not reimbursed, people won’t pay for it, that’s clear. It’s obvious—it’s already unbelievable how much we pay for insurance (...) It takes a lot of time to talk about food, to hear about these problems. But how can he do it? Unless he takes an hour. Not with the little time they have, that’s not possible, is it? It could be to my detriment.*”(P01)

From the GPs’ perspective, the health financing scheme limits their ability to conduct thorough medical consultations that include preventive measures. They were therefore sceptical about their capacity to integrate environmental aspects:

“*It’s mainly a question of time to have a full discussion on these issues (...) it can be a bit overwhelming, especially in terms of time. And as a result, it doesn’t allow us to address other health issues the patient came for.*”(GP04)

Despite the time constraints and the need to preserve a trustful, judgement-free relationship with patients, some participants noted that GPs could still subtly introduce environmental topics alongside health advice, as suggested by this patient:

“*The dominant role, which is that of health, must remain dominant in my opinion. But on the other hand, having a second layer... that can be introduced as a complement to a health recommendation—I think that could be well received...*”(P02)

GPs also reported that environmental considerations could be conveyed indirectly, through the setting of their practice, their behaviour, or subtly expressed values:

“*I think we can give advice by cycling to do a home visit.*”(GP05)

“*Values ... that’s what I put on the wall... that’s my bike helmet hanging there. It’s (...) people can see that I don’t drive a 4-wheel drive.*”(GP02)

### 3.3. How to Better Tailor the Message?

Following the concerns about legitimacy, time constraints, and the risk of moralising or politicising climate-related messages in clinical settings, some participants who supported a co-benefits approach offered suggestions on how to better tailor the communication strategies. Several participants emphasised the importance of promoting preventive messages about health/environmental co-benefits, while avoiding guilt-inducing or moralising “activist” tones. Instead, they recommended that informational materials rely on factual and quantitative data, endorsed by scientific and medical authorities.

One participant stressed the importance of structuring the information in a way that follows a chronological narrative, allowing individuals to relate to the content and feel involved. He added that helping people understand how their actions—when multiplied across a community—can have a meaningful impact and fosters a sense of belonging:

“*In communication, the chronology of information is quite important because if you give information too early, you can’t link it to other information (…) It’s to be linked, perhaps, with the two slices of meat per week that we agree to eat and the five others that we give up. If we multiply the five slices by eight million people in Switzerland, that’s a lot!*”(P02)

Echoing earlier concerns about judgement and guilt, most participants stressed that communication should be positive and constructive, encouraging lifestyle changes by focusing on personal benefits and the pleasure associated with the proposed change:

“*All messages with deprivation do not work. Fear doesn’t work either. You have to encourage and not criticize.*”(PE)

Some GPs also shared practical ideas for subtly introducing environmental considerations into clinical settings. One GP mentioned giving concrete advice, such as biking to avoid traffic, while another suggested displaying vegetarian cookbooks in the waiting room to inspire change indirectly:

“*Another possibility is to put beautiful vegetarian cookbooks in our waiting rooms. Indirectly, it’s not talking about the climate, but it’s making people want to eat vegetarian recipes.*”(GP01)

## 4. Discussion

The potential role of health professionals in raising awareness on climate change and implementing mitigation strategies through the lens of co-benefits in clinical settings has been discussed in several publications. For example, a study among GPs in Switzerland showed their willingness to address health and environment co-benefits with patients, while identifying lack of time, guidelines, and knowledge as key barriers to integrating this topic into clinical practice [8].

This exploratory project aimed to address some of these barriers by providing informational materials—posters and factsheets—and assessing their understanding and acceptability in GP practices. By collecting the view of both patients and GPs engaging with these materials, our study reveals additional barriers, including personal positions that were often conflicting and ambivalent.

While André et al. demonstrated a generally favourable attitude toward integrating co-benefits into clinical practice [8], our findings show that, in practice, GP engagement was more equivocal. This study builds on André et al.’s work [8] by incorporating patients’ perspectives and offering insight into how co-benefits messaging is received and interpreted in practice.

Importantly, participants suggested potential facilitators for improving message reception, such as promoting positive, non-moralising messages with a collective perspective and tailoring them to individual lifestyles. These suggestions point to the need for more nuanced and context-sensitive communication strategies.

The patients and GPs acknowledged the pertinence of linking individual health with environmental health, particularly in relation to diet and mobility. At the discourse level, there was general agreement that receiving information from GPs on these topics was appropriate. However, some patients reacted negatively to messages perceived as guilt-inducing or overly negative, which triggered feelings of powerlessness. The topic of reducing meat consumption, and its implications for individual health, was particularly debated. Both patients and GPs highlighted conflicting dietary recommendations—such as those from the EAT-Lancet Commission versus the *Société Suisse de Nutrition* [Swiss Society of Nutrition]—which created confusion and undermined confidence in the reliability of the evidence. Delorme et al.’s study reports these similar disparities, which were induced by the consideration of environmental factors in healthy diet prescriptions [23]. This ambiguity reflects broader tensions in scientific knowledge production and the challenge of navigating competing recommendations. It also underscores the potential role of GPs in clarifying evidence-based guidance on diet and health.

Our findings suggest that health-related behaviour change messages should be tailored to patients’ socioeconomic contexts. For example, the accessibility—both physical and financial—of healthy and sustainable food must be considered. A recent study on sustainable food prescriptions in Switzerland emphasised how daily life factors such as time, mobility, and social relationships shape eating habits [24]. These insights support the idea that co-benefits messaging is best delivered through personalised, face-to-face dialogue.

However, consistent with previous research, we found that the consultation setting is perceived as constrained by time and billing schemes, making it an unsuitable space for extended discussions on co-benefits. From the patients’ perspective, this limits the feasibility of integrating such topics into routine care.

The reactance expressed by GPs reflects a form of self-censorship, driven by concerns that addressing climate change—a topic perceived as political—might compromise the trust-based relationship with patients. Although all participating GPs had agreed to display thematic materials (posters and factsheets) in their waiting room, many expressed discomfort with discussing environmental health directly during consultations. These topics were seen as potentially intrusive, touching on patients’ private sphere and freedom, and thus risking the therapeutic alliance.

This finding was unexpected. We had assumed that participating GPs were committed to addressing the health consequences of climate change in clinical encounters. Moreover, discussing lifestyle habits such as diet and physical activity is typically part of routine preventive care in general medicine. Yet, when these habits were framed in relation to climate change, they were perceived as more intrusive, politically charged, and problematic. This suggests that the environmental framing introduces a layer of moral and political complexity that alters the perceived legitimacy of the discussion.

In Switzerland, health discourse is deeply embedded in a rationale of individual responsibility, where individuals are expected to manage their own health through personal choices and behaviours—how they eat, move, and consume throughout their life [25]. However, some GPs were reluctant to extend this logic to climate-related behaviours, fearing that it would induce guilt or be perceived as judgmental and moralistic. This highlights a tension between public health goals and the preservation of a non-judgmental clinical space.

The GPs’ complex positionality also emerged in their reflections on the perceived divide between the ‘political’ and the ‘scientific’. Several GPs viewed climate discussions as inherently political, aligning with personal convictions they preferred not to express in a professional setting. Instead, they emphasised a desire to “remain scientific”, suggesting that medical practice should be grounded in science and neutrality. We situate this position in the specific context of the Swiss healthcare system, where GPs operate as liberal professionals in private practices, often detached from broader public health frameworks.

This positioning vis-à-vis public health matters contrasts with historical views such as Virchow’s assertion that “medicine is a social science, and politics nothing but medicine at a larger scale” [26]. In this light, the reluctance of GPs to engage with the environmental topic may stem from their limited identification with public health advocacy, a role that remains underdeveloped in Swiss primary care [27].

The organisation of the Swiss healthcare system further reinforces this individualistic orientation. Most GPs work in small, fee-for-service practices, which limits their capacity and incentive to engage with systemic or collective health issues [28]. This structural context contributes to the perception that environmental health is beyond the scope of routine medical care.

Patients’ reactions to the posters—particularly those addressing meat consumption—also revealed tensions. Many perceived the messages as unduly moralistic, controversial, and overly focused on individual responsibility. This suggests that ecological messaging can provoke emotional and psychological reactions and resistance, especially when it conflicts with dominant societal narratives or personal values.

Interestingly, while patients in Switzerland are accustomed to health messages emphasising individual responsibility in health matters, they appear less inclined to apply this logic to climate change. Many pointed to the greater responsibility of industry and institutions, questioning the fairness of messages that target individual behaviour. This disconnect between global issues—the environment—and personal health choices underscores the need for clearer, more relatable messaging.

The ambiguity of the messages—especially regarding dietary recommendations—further complicated their reception. Conflicting guidelines from different authorities (e.g., EAT-Lancet vs. Swiss Society of Nutrition) created confusion and undermined trust in the scientific basis of the advice. This highlights the importance of clarity, consistency, and credible sources in the visual and textual design of informational materials.

Patients’ reactance should also be interpreted within the broader complex context of contemporary capitalist societies, where consumer habits are deeply ingrained and often reinforced by powerful economic actors. The influence of agroeconomic and pharmaceutical industries—largely present and influential in the Swiss context—in shaping public discourse and policy contributes to a general sense of public distrust [29,30,31]. Recommendations to reduce meat consumption or promote cycling, for example, challenge dominant norms and may be perceived as ideological rather than health-driven.

This study underscores the necessity of formulating materials that are aligned, as much as possible, with the diverse opinions surrounding climate change. The content and visual design of the posters provoked negative responses among those less familiar with, or less convinced by, climate-related issues. It therefore failed to engage them in a discussion about co-benefits. In this sense, this study revealed shortcomings in both visual design and messaging, which could have been mitigated through a stronger co-production strategy. However, this does not diminish the finding that GPs were limited in their ability to address such topics with patients.

These findings call for a deeper reflection on the concept of health as a social good and right in democratic societies like Switzerland. They also point to the need for structural changes in the medical system, including a re-evaluation of the rationalist paradigm of biomedicine. Integrating a patient–planetary health approach into clinical practice would support GPs in raising awareness about the links between health and the environment [32].

This study also highlights several possible avenues for integrating a co-benefits perspective in primary care. Some of the suggestions below emerged during the FGD conducted with participating GPs after presenting the main findings of this study.

GPs can play a role in clarifying points of confusion, such as the absence of negative health impacts from reducing meat consumption when adopting sustainable diet recommendations like those of the EAT Lancet Commission [19]. This information can easily be adapted for patients with specific needs (e.g., chronic diseases, anaemia, intense sport activities) and can be made easily accessible to GPs.GPs can play a key role in helping patients navigate the reliability of health information in societies strongly shaped by neoliberal and market-driven influences. Some diet recommendations—particularly those related to reducing animal-based foods or sugar—may be shaped by agroeconomic interests [29,31], contributing to public confusion. GPs are well-positioned to act as trusted intermediaries, helping patients interpret and contextualise such information.Since discussing health habits is already part of routine medical practice, in relation to primary and secondary prevention, GPs can introduce co-benefits without explicitly referencing environmental health. By focusing on individual health benefits, they can avoid triggering political or moral resistance while still promoting sustainable behaviours.GPs and practice staff can act as subtle role models through the visual and symbolic environment of medical practice. For example, displaying vegetarian or low carbon cookbooks in the waiting room, or visibly using active transport (e.g., cycling to home visits), can communicate values without direct verbal engagement. These visual cues can help normalise co-benefits behaviours and reinforce the message without moralising.The visual design and framing of informational materials should be rethought. Messages should emphasise the collective impact of individual actions, helping patients feel part of a broader effort rather than isolated or powerless. Materials should also be adapted to patients’ socioeconomic position, cultural background, and living circumstances, to ensure relevance and avoid alienation. This would help reduce the ambiguity and emotional resistance observed in this study.Better tools and educational materials are needed for clinical practice. These should be co-designed with both patients and GPs—to ensure usability and acceptability. GPs need clear, referenced guidelines and tailored recommendations to feel confident introducing these topics in a way that aligns with patients’ specific needs. This could also reduce patients’ perception of the information as unreliable or controversial. Furthermore, training on patient–planetary health co-benefits should be integrated into GP education to strengthen their knowledge, communication skills, and sense of legitimacy in addressing these issue with patients [32].This study is part of a broader initiative to reform medical education worldwide and specifically at the Faculty of Biology and Medicine of the University of Lausanne. The integration of sustainability and environmental knowledge such as health/environment co-benefits approaches, both for clinical care practices and healthcare reforms, is indeed expected to lead to more concrete and actionable recommendations for medical curricula. In that perspective, the present study aims to better define how to integrate co-benefits related to mobility and diet into general practice.

### Strengths and Limitations

A key strength of this study is that it gave voice to both patients and GPs on the topic of discussing co-benefits in medical settings. It enabled revealing the difficulty GPs face in positioning themselves in a dualistic perception of the private vs. public and the scientific vs. political—dimensions that had not emerged through survey data [8]. It also included patients from diverse walks of life, as they were recruited and invited to speak in vivo in their GP’s waiting room. Such diversity is not often captured in surveys due to recruitment biases.

The study also has limitations. The findings must be contextualised within the specific setting of general medicine practice in private clinics in Western Switzerland and may not be generalizable to other settings. As stated, general medicine practices in Switzerland are private, and although health insurance is compulsory to ensure universal coverage, patients face varying premiums and co-payments levels. The limited acceptance of co-benefits discussions during consultations—partly paid out of pocket—may be specific to this health financing scheme and may not apply to other settings.

The sample of interviewed GPs and patients was limited in number, and the diversity of GPs’ perspectives may be constrained by selection bias. Participants’ profiles were not collected systematically during interviews, limiting the exploration of how sociodemographic positions influenced viewpoints. Finally, participants’ awareness of the study’s environmental focus may have influenced their responses, potentially introducing desirability bias. Individuals with sceptical views on climate change may be underrepresented.

## 5. Conclusions

The present study illustrates the complexity of addressing climate change and environmental issues within medical practice. Discussions with patients and GPs revealed ambivalent attitudes and varying levels of knowledge about climate change. Notably, it also highlighted different perceptions of the roles GPs are expected to play. In relation to climate change, GPs expressed the need to position themselves within dualist frameworks—such as science vs. politics, or individual vs. society. The findings also reveal the blurred boundary between what is considered “political” and what is not in the medical field, a boundary that is likely shaped by broader societal context. This underscores the importance of understanding how current social narratives influence perceptions of legitimacy in clinical discussions.

These insights point to the need for interdisciplinary approaches in clinical practice to develop and evaluate innovative tools that promote dialogue on the links between health and the environment. Further research should consider drawing on behavioural theory and exploring structural changes needed to support transitions toward increased co-benefits. This includes considering how environmental issues can be effectively integrated into the concerns of public health and the medical system.

## Figures and Tables

**Table 1 ijerph-22-01503-t001:** Participants’ profiles.

Patients’ Profiles’ Information					General Practitioners’ Profiles	
Observations			Interviews			
Type of Medical Practice	Number of Patients	Information on Patients’ Profiles	Information on Patients’ Profiles			
A. Urban medical practice (canton de Neuchâtel)	8 women and 3 men	The observed patients tended to be older.	P03	A woman in her forties from Eastern Africa, works as a housecleaner.	GP01	A 50-year-old woman practicing as a family physician.
			P04	A woman in her sixties with a university degree.		
			P05	A man in his thirties with a GP partner.		
B. Urban medical practice (canton de Vaud)	5 women and 5 men	The observed patients were diverse in age and multicultural.			GP03	A 47-year-old woman practicing as a family physician with additional qualifications in acupuncture and Chinese pharmacotherapy.
					GP04	A 57-year-old man practicing as a family physician with additional expertise in infectious diseases, tropical medicine, and travel medicine.
C. Urban medical practice (canton de Vaud)	7 women and 1 man	The observed patients were predominantly female, with an age range of 40 to 80 years, and primarily sought acupuncture treatment.				
D. Urban medical practice (canton de Vaud)	3 women and 2 men	The clientele was, with an age range of approximately 40 to 80 years.	P01	A retired nurse and humanitarian aid worker from France.	GP02	A 51-year-old man practicing as a family physician.
E. Medical practice in rural area (canton de Vaud)	4 women and 3 men	The clientele comprised individuals between the ages of 40 and 85.	P02	A man in his sixties from Italy, working in communications.	GP05	A 54-year-old woman practicing as a family physician.

## Data Availability

The data used are confidential.

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
