# Peer review of "Discussing Human and Environmental Health Co-Benefits Related to Diet and Mobility Behaviours in the Primary Care Setting: A Qualitative Exploratory Study"

_ijerph, 2025, doi:10.3390/ijerph22101503_

Round 1
Reviewer 1 Report
Comments and Suggestions for Authors
Thank you for opportunity to review your interesting manuscript.
General assessment:
This manuscript explores a timely topic at the intersection of environmental sustainability, and clinical communication. The study is well-motivated, methodologically appropriate for its exploratory aims, and makes a valuable contribution to understanding how climate-related health communication might be operationalized in clinical practice.
Study's originality and relevance:
The manuscript addresses a gap in the literature by integrating both patient and GP perspectives on environmental health messaging in clinical settings. This dual perspective adds to the discussion.
Contributions:
I think that this study makes meaningful contribution to its field. It highlights the complexity patients experience when exposed to environmental health information during clinical encounters. Importantly, the paper offers practical suggestions for message delivery, including subtle modeling of sustainable behaviors and non-intrusive educational cues. Contributions are valuable given the growing interest(and need) in integrating climate action into healthcare practice.
Methodological considerations:
The qualitative design is appropriate for the study’s exploratory nature, and the use of observations adds depth.
- Coding and thematic analysis process would benefit from more detail, particularly around steps taken to ensure consistency or intercoder reliability.
- It would be useful to explain how the waiting room materials were developed and whether any pre-testing or pilot evaluation was conducted.
- Participant awareness of the study’s environmental focus could have influenced responses; future work might consider more neutral framing to reduce response bias.
Validity of conclusions:
The conclusions are logically derived from the findings and well supported by the data. Emphasis on health-centered framing, non-judgmental tone, and subtle visual cues is well grounded in the qualitative evidence. The recommendation for increased GP training is also appropriately justified.
References:
The manuscript is well supported by current and relevant literature. It draws on key international and national sources to situate the study in both local and international contexts. One possible enhancement would be to integrate behavioral theory frameworks to help interpret patient responses and resistance.
Presentation:
The inclusion of visual elements like a diagram of the study design, a summary table of thematic findings, or a model of GP-patient interaction pathways would improve clarity. The writing is clear, structured, and appropriate for an academic readership.
Ethical Considerations:
The authors demonstrate commendable reflexivity in acknowledging their role in developing the intervention and the potential for bias. Ethical procedures are clearly outlined and appropriately followed.
Suggestions:
- Consider expanding the discussion to include cross-cultural perspectives or comparisons to other healthcare systems to situate the findings more broadly.
- The manuscript would benefit from more explicit recommendations for policy or clinical training, particularly regarding how co-benefits messaging might be integrated into medical education or supported through healthcare system incentives.
Recommendation:
Minor revisions suggested to enhance methodological transparency, presentation of findings, and theoretical framework.
Author Response
Comment 1 : Coding and thematic analysis process would benefit from more detail, particularly around steps taken to ensure consistency or intercoder reliability.
Response 1: Thanks for your comment. We have clarified the analytical process in the section 2.5 Data Analysis. We have described the different stages of the analysis in greater detail.
Section 2.5 Data analysis (page 6, lines 205-206):
Data were analysed by AS and ER using thematic analysis [21]. Conducted as an iterative and continuous process, the initial analysis informed subsequent data collection and further analytical steps. The analytical process began with the observation notes, which constituted the initial dataset. AS and ER jointly analysed these notes, as they had conducted the observations together. They collaboratively identified salient features, coded relevant segments, and collated them into preliminary themes. These themes were then discussed and refined with the other two authors (NS, a male physician and epidemiologist, and JS, a female sociologist and epidemiologist). They were subsequently used to guide further analysis during the individual interviews. In a second phase, the interview transcripts were fully analysed separately by AS and ER. Coding was conducted using the previously identified themes, while allowing for the inductive emergence of new themes. The new and final themes were compared and discussed in conjunction with NS and JS. The final FGD was used to reach consensus on the final thematic map and ensure analytical coherence. Elements brought by FGD participants were used to develop the recommendations presented in this article. Finally, the entire dataset was comprehensively reviewed using the final thematic map, with a triangulation of findings from observation, semi-structured interviews, and FGD. The analytical process was regularly discussed within the team to maintain coherence and consistency.
Comment 2 : It would be useful to explain how the waiting room materials were developed and whether any pre-testing or pilot evaluation was conducted.
Response 2: Thank you for pointing this out. We agree that a pre-testing/pilot evaluation would have been useful, but unfortunately the time available for the study did not allow us to conduct a pilot phase. However, the information materials were discussed during working sessions with the team, including health professionals and a patient, to define the most suitable format for a waiting room. The content and visuals were refined several times. We have added more details in the section 2.2 study material.
Section 2.2 Study material now reads as follows (page 3, lines 122-143):
This study was conducted through a partnerial approach involving a university institution and a non-governmental organisation (Engagés pour la santé). Accordingly, the research material was developed in close collaboration with the local healthcare association. The co-benefits material was developed by the project team (authors of this paper, Julia Gonzales Holguera, and four health professionals from the Swiss association Engagés pour la Santé [Committed to Health], namely two GPs, one physiotherapist and patient, and one psychiatrist, all women). Collaborative working sessions were held to define the focused co-benefits topics and the optimal format for an intervention in a waiting room. The team decided to focus on two topics: meat consumption and active mobility; and to develop two types of support material: posters and a factsheet (the study material is available in the Supplementary Materials). The two posters, one for each topic, were created using existing material developed by local healthcare professionals’ associations—Engagés pour la Santé and Médecins en faveur de l’environnement [Doctors for the environment]—involved in co-benefits approaches. During the working sessions, the team discussed the posters' content and visuals, which were adapted and reviewed at the following working session, until they reached a consensus. The team modified the content to avoid guilt-inducing messaging and to ensure design consistency with the other project materials. The factsheet used the same existing material, supplemented with additional information from the EAT-Lancet Commission data [19].
Comment 3 : Participant awareness of the study’s environmental focus could have influenced responses; future work might consider more neutral framing to reduce response bias.
Response 3 : Thank you for your comment; we acknowledge it and will take it into consideration for future studies. We have added it in the study’s limitations.
Page 20, paragraph 1, lines 817-821: Finally, participants’ awareness of the study's environmental focus may have influenced their responses, potentially introducing desirability bias. Individuals with sceptical views on climate change may be underrepresented.
Comment 4 : The manuscript is well supported by current and relevant literature. It draws on key international and national sources to situate the study in both local and international contexts. One possible enhancement would be to integrate behavioral theory frameworks to help interpret patient responses and resistance
Response 4: Thank you for your comment. We acknowledge that integrating a behavioural theory framework could improve the study, and we will take this into consideration for future studies. Although the team was interdisciplinary, we lacked expertise in behavioural theory. We pointed this in the conclusion (page 20, paragraph 2, lines 838-842) : “These insights point to the need for interdisciplinary approaches in clinical practice to develop and evaluate innovative tools that promote dialogue on the links between health and the environment. Further research should consider drawing on behavioural theory and exploring structural changes needed to support transitions toward increased co-benefit.”
Comment 5 : The inclusion of visual elements like a diagram of the study design, a summary table of thematic findings, or a model of GP-patient interaction pathways would improve clarity. The writing is clear, structured, and appropriate for an academic readership.
Response 5 : Thank you for your comment. We have added a table of participants’ profiles at page 5, line 173.
Table 1 : Participants’ profiles
|
Patients’ profiles’ information |
General Practitioners’ profiles |
|||||
|
Observations |
Interviews |
|||||
|
Type of medical practice |
Number of patients |
Information on patients’ profiles |
Information on patients’ profiles |
|||
|
A. Urban medical practice (canton de Neuchâtel) |
8 women and 3 men |
The observed patients tended to be older. |
P03 |
A woman in her forties from Eastern Africa, works as a housecleaner. |
GP01 |
A 50-year-old woman practicing as a family physician. |
|
P04 |
A woman in her sixties with a university degree in Literature. |
|||||
|
P05 |
A man in his thirties with a GP partner. |
|||||
|
B. Urban medical practice (canton de Vaud) |
5 women and 5 men |
The observed patients were diverse in age and multicultural. |
|
|
GP03 |
A 47-year-old woman practicing as a family physician with additional qualifications in acupuncture and Chinese pharmacotherapy. |
|
GP04 |
A 57-year-old man practicing as a family physician with additional expertise in infectious diseases, tropical medicine, and travel medicine. |
|||||
|
C. Urban medical practice (canton de Vaud) |
7 women and 1 man |
The observed patients were predominantly female, with an age range of 40 to 80 years, and primarily sought acupuncture treatment. |
|
|
|
|
|
D. Urban medical practice (canton de Vaud) |
2 women and 3 men |
The clientele was, with an age range of approximately 40 to 80 years. |
P01 |
A retired nurse and humanitarian aid worker from France. |
GP02 |
A 51-year-old man practicing as a family physician. |
|
E. Medical practice in rural area (canton de Vaud) |
4 women and 3 men |
The clientele was comprised of individuals between the ages of 40 and 85. |
P02 |
A man in his sixties from Italy, working in communications. |
GP05 |
A 54-year-old woman practicing as a family physician. |
Comment 6 : Consider expanding the discussion to include cross-cultural perspectives or comparisons to other healthcare systems to situate the findings more broadly
Response 6 : This is a good point. We had already situated our research into the specific context of Switzerland:
Page 16, paragraph 2, lines 630-637: In Switzerland, health discourse is deeply embedded in a rationale of individual responsibility, where individuals are expected to manage their own health through personal choices and behaviours - how they eat, move, and consume throughout their life [25].
Page 16, paragraph 4, lines 652-656: We situate this position in the specific context of the Swiss healthcare system, where GPs operate as liberal professionals in private practices, often detached from broader public health frameworks.
We have however expanded the discussion, with the new “Strengths and limitations” section, discussing how findings from Switzerland are specific to the financial scheme of the health system and thus may not be generalisable to other settings.
Section 4.1, page 19, paragraph 2, lines 807-813: The study also has limitations. The findings must be contextualized within the specific setting of general medicine practice in private clinics in Western Switzerland and may not be generalizable to other settings. As stated, general medicine practices in Switzerland are private, and although health insurance is compulsory to ensure universal coverage, patients face varying premiums and co-payments levels. The limited acceptance of co-benefits discussions during consultations – partly paid out-of-pockets – may be specific to this health financing scheme and may not apply to other settings.
Comment 7 : The manuscript would benefit from more explicit recommendations for policy or clinical training, particularly regarding how co-benefits messaging might be integrated into medical education or supported through healthcare system incentives
Response 7 : We thanks your for this suggestion. We agree with this point. Therefore, we have added more explicit recommendation in the discussion.
Page 19, paragraph 2, lines 790-796 : This study is part of a broader initiative to reform medical education worldwide and specifically at the Faculty of Biology and Medicine of the University of Lausanne. The integration of sustainability and environmental knowledge such as health/environment co-benefits approaches, both for clinical care practices and healthcare reforms, is indeed expected to lead to more concrete and actionable recommendations for medical curricula. In that perspective, the present study aims to better define how to integrate co-benefits related to mobility and diet into general practice.
Comment 8 : Minor revisions suggested to enhance methodological transparency, presentation of findings, and theoretical framework
Response 8 : Thank you very much for taking the time to review our manuscript and for your relevant feedback. We hope that our changes to the manuscript meet your expectations.
Reviewer 2 Report
Comments and Suggestions for Authors
Dear authors,
it was a pleasure to read this paper. I propose a minor change with a few suggestions. I believe that they will not take up too much of your time.
- The authors state that they want to investigate "patient and physician receptivity", but it would be useful to specify what specifically is new in this research compared to the previous work of André et al. (2022).
- Although the term "co-benefits" is used, it is loosely defined in my opinion. It is very important and deserves a separate chapter.
- At the very end of the introduction, the aim of the study is described quite descriptively. It is recommended that the objectives be expressed more precisely.
- The method suggests insufficient transparency of sampling. In which districts was the investigation conducted? How were patients selected? How many patients dropped out?
- To increase transparency, try to describe your participants in more detail in a summary table.
- Also, consider including the posters in the supplement. Thus, readers could connect their thoughts about the messages on the posters with their content.
- Have you used any software tools in the analysis?
- Throughout the results and discussion, the line between participants' voices and authors' interpretation is often blurred. It is difficult to distinguish findings from interpretation.
- It is not clear enough why physical activity was not noticed — was it because of the visuals, the familiarity of the message, or the unrecognizable language? It remains unanalyzed.
- Consider using numbers instead of just P and GP. For example, P01, GP04... Maybe readers will connect the thoughts of the participants.
- Chapter 3.3 needs to be elaborated in more detail. Try to connect it with earlier chapters.
- There is insufficient insight into the difference by socio-demographic groups. Answers are not analyzed: who makes which suggestions? Do younger respondents suggest digital messages and older respondents suggest informational flyers? Do women suggest more emotional messages? That would enrich the conclusions about the creation of messages according to target groups.
- The financial dimension is mentioned but sparingly. The question of who is going to pay for the additional part of the consultation is extremely important — but not elaborated enough.
- Consider an explanation of the broader framework in which primary health care is implemented. I just read in the discussion that it is private in Switzerland. This way of organizing presents a special challenge for the implementation of such campaigns.
- State the limitations of this research.
- The discussion should be better organized. The authors mention terms such as "ambivalence" and "political perception", but do not elaborate on them conceptually.
- There is a lack of deeper reflection on mechanisms of influence: Why would positive messages work more? Does it have anything to do with cognitive processing, motivation, identity? It is possible to refer here to theories of health communication.
- Failures of the intervention are not explicitly discussed. The intervention obviously did not achieve the main goal — recognition of co-benefit messages and their spontaneous introduction into the conversation. The authors relativize it. Suggestion: openly admit design failures and include reflection on visual design, ambiguity of messages, and unpreparedness of GPs for these topics.
Of course, this work has a lot of value that I have not mentioned here, but which every careful reader will come across. It seems to me that both doctors and patients clearly assess the difficulties, limitations, and their possibilities.
Comments on the Quality of English LanguageAbove all, the text should be changed to make it easier to read. Enter chapters. Visualization. Also stick to the good old approach of one paragraph dealing with one topic. Where possible, shorten the text and avoid excessive elaboration. The authors have quite enough material that they should present.
Author Response
Comment 1 : The authors state that they want to investigate "patient and physician receptivity", but it would be useful to specify what specifically is new in this research compared to the previous work of André et al. (2022).
Response 1: Thank you for reviewing our manuscript and for your valuable comments. To better situate this study, that indeed builds on previous work from our group we have brought some modifications to the manuscript. This study builds on the previous André et al. study, and has three specificities: (1) it assesses the physician perceptions in real-world setting (vs in theory in André’s study); (2) it assesses their perceptions qualitatively, enabling in-depth understanding of the rationale for a favourable or unfavourable opinion; and (3) it integrated the voice of patients. We had described in the introduction and the discussion how this research is in continuity with André et al’s paper. We have made some modifications in the introduction and the discussion sections, that now read as follows (reviewed items are in red):
Section 1. Introduction Page 2-3, paragraph 3, lines 82-106 :
A study included in the systematic review was conducted by André et al in Switzerland [8]. It explored general practitioners’ (GP) knowledge and attitudes on climate change and health, and their interest in promoting a co-benefit approach in their clinical practice with patients. The study demonstrated that GPs believe patients should be better informed about the links between global warming and health, that it could be part of the medical consultation and that some were already discussing climate change with their patients. Corroborating with other existing evidence, the perceived barriers were lack of time, of scientific recommendations or even of knowledge. The study also suggested that the political orientation of GPs played a role in their willingness to adopt a co-benefits approach.
Following-up on the study from André et al, and responding to the perceived need for educational material, a project was launched in September 2021 in GP practices of Western Switzerland that aimed at developing material related to co-benefits to be displayed and used in GP practices. By displaying the material in waiting rooms, the intervention objective was to trigger interest in patients, discussions between patients, and an opportunity for GPs to discuss the topic of meat consumption and active mobility with patients during the consultation. Going beyond the study on knowledge and attitudes of André et al, this study assessed the implementation of a co-benefits approach in real-world practice. The aim of this study was to qualitatively assess the implementation of a co-benefits approach in real-world practice. We specifically aimed at assessing (i) the understanding and perception of the co-benefits’ messages from the point of view of patients; and (ii) to examine GPs' perceptions of their role and legitimacy in addressing co-benefits with patients during a clinical encounter.
Section 4. Discussion Page 14, paragraph 3, lines 555-560:
While André et al. demonstrated a generally favorable attitude toward integrating co-benefits into clinical practice, our findings show that, in practice, GP’s engagement was more equivocal. This study builds on André et al.’s work by incorporating patients’ perspectives, and offering insight into how co-benefits messaging is received and interpreted in practice.
Importantly, participants suggested potential facilitators for improving message reception, such as promoting positive, non-moralizing messages with a collective perspective and tailoring them to individual lifestyles.
Comment 2 : Although the term "co-benefits" is used, it is loosely defined in my opinion. It is very important and deserves a separate chapter.
Response 2: Thank you for your comment. We have expanded on the definition of this concept in the introduction, and we have included an additional reference to strengthen the conceptual understanding of co-benefits.
Page 2, paragraph 1, lines 51-60 :
Regarding this latter role, studies have highlighted the urgency for action and the important contribution that health professionals can make in disseminating information and raising awareness among populations and health policy makers [5,8,10–15]. The ability of health professionals to engage with individuals and promote preventive behavioural changes, such as healthy habits (e.g., smoking cessation) is viewed as a valuable skill that professionals can use in their clinical practice. It enables them to influence patients’ behaviours in a way that benefit both the environment and human health [8,12,16,17]. To support this approach, the concept of co-benefits is employed to emphasis the reciprocal advantages for human health and ecosystems that result from strengthening the connections between environment and health. This concept is defined as the positive health outcomes that emerge from interventions aimed at mitigating environmental degradation or, conversely, from initiatives that promote environmental preservation while simultaneously enhancing human health [12]. Among others, Hopkins and al contributed to a better understanding of the concept of co-benefits, which they referred as “win-win interventions” in their publication [18].
Comment 3 : At the very end of the introduction, the aim of the study is described quite descriptively. It is recommended that the objectives be expressed more precisely.
Response 3 : We agree with your comment and we specified the aim of the study at the end of the introduction. We clarified the objectives of the intervention vs the aim of the study.
Page 3, paragraph 1, lines 100-106:
The aim of this study was to qualitatively assess the implementation of a co-benefits approach in real-world practice. We specifically aimed at assessing (i) the understanding and perception of the co-benefits’ messages from the point of view of patients; and (ii) to examine GPs’ perceptions of their role and legitimacy in addressing co-benefits with patients during a clinical encounter.
Comment 4 : The method suggests insufficient transparency of sampling. In which districts was the investigation conducted? How were patients selected? How many patients dropped out?
Response 4: We have taken your comment into account and specified the number of Cantons and added the names of the Cantons in which the study was conducted. We have also specified how patients were recruited in waiting rooms and the number who refused to participate in the study. Regarding the interviews, all participants who showed interest and were available for the interview phase took part in the study. As the interviews were scheduled directly during the observations, there were no withdrawals. The sampling procedure section has been modified, as follows (changes are in red):
Section 2.3 Sampling Procedure (page 4, lines 149-165):
Family medicine practices from Western, francophone Switzerland were contacted by email and invited to take part into the study, presented as a follow-up of André et al’s study. Practices were selected from a list of the Department of Family Medicine (DFM) that groups GPs collaborating with the department for education, clinical supervision, or research activities. A total of 65 practices were contacted across 4 Cantons (Vaud, Neuchâtel, Jura and Genève). Eight practices responded, expressing interest in participating in the study. However, 2 practices eventually withdrew during the phase of the observation (as discussed in the Result section). The study was therefore conducted in six practices located in the Cantons of Vaud and Neuchâtel. Medical practices were thus recruited using a convenience sampling procedure (see Table 1 for a description of medical practices). Patients attending consultation days were included in the study. During the observations, all patients present in the waiting rooms were invited to share their reaction to the information material. In each medical office, a small number of patients (between 1 and 3) declined to participate in the discussion. Patients who expressed interest in further discussing the topic were recruited to pursue with individual interviews to be conducted in their preferred location—their home or the DFM office. Many declined due to lack of interest or of time. To complement patient recruitment, one participating GP recruited and organised 3 patient interviews, which were conducted at their practice. Regarding GPs interviews, all participating GPs were invited to take part in an individual interview, as well as to a final FGD to collect their experience and perceptions of the project. One GPs was unavailable for the interview.
Comment 5 : To increase transparency, try to describe your participants in more detail in a summary table.
Response 5 : Thank you for your comment. Due to the specific nature of the medical waiting room context, it was not possible to collect the personal information of participants during the observations. Some information was collected during the interviews, but not systematically. We acknowledge this limitation of our study and have included it in our paper. We have also added a summary table of participants' profiles in section 2.3.
page 5, line 173: Table 1 : Participants’ profiles
|
Patients’ profiles’ information |
General Practitioners’ profiles |
|||||
|
Observations |
Interviews |
|||||
|
Type of medical practice |
Number of patients |
Information on patients’ profiles |
Information on patients’ profiles |
|||
|
A. Urban medical practice (canton de Neuchâtel) |
8 women and 3 men |
The observed patients tended to be older. |
P03 |
A woman in her forties from Eastern Africa, works as a housecleaner. |
GP01 |
A 50-year-old woman practicing as a family physician. |
|
P04 |
A woman in her sixties with a university degree in Literature. |
|||||
|
P05 |
A man in his thirties with a GP partner. |
|||||
|
B. Urban medical practice (canton de Vaud) |
5 women and 5 men |
The observed patients were diverse in age and multicultural. |
|
|
GP03 |
A 47-year-old woman practicing as a family physician with additional qualifications in acupuncture and Chinese pharmacotherapy. |
|
GP04 |
A 57-year-old man practicing as a family physician with additional expertise in infectious diseases, tropical medicine, and travel medicine. |
|||||
|
C. Urban medical practice (canton de Vaud) |
7 women and 1 man |
The observed patients were predominantly female, with an age range of 40 to 80 years, and primarily sought acupuncture treatment. |
|
|
|
|
|
D. Urban medical practice (canton de Vaud) |
2 women and 3 men |
The clientele was, with an age range of approximately 40 to 80 years. |
P01 |
A retired nurse and humanitarian aid worker from France. |
GP02 |
A 51-year-old man practicing as a family physician. |
|
E. Medical practice in rural area (canton de Vaud) |
4 women and 3 men |
The clientele was comprised of individuals between the ages of 40 and 85. |
P02 |
A man in his sixties from Italy, working in communications. |
GP05 |
A 54-year-old woman practicing as a family physician. |
Comment 6 : Also, consider including the posters in the supplement. Thus, readers could connect their thoughts about the messages on the posters with their content.
Response 6 : We agree and have added the information material in the Supplementary Materials.
Comment 7 : Have you used any software tools in the analysis?
Response 7 : No. We did not use any software for the analysis. Instead, we manually coded the transcripts and observation notes in Word documents, applying colour coding and comments. We compiled emerging themes in separate Word files, along with coded excerpts from the interviews, to provide an overview.
Comment 8 : Throughout the results and discussion, the line between participants' voices and authors' interpretation is often blurred. It is difficult to distinguish findings from interpretation.
Response 8 : We have fully revised the results and discussion sections with a focused attention to this point, in addition to English proofing, and to ensure the 1 paragraph – 1 idea structure. We have modified our writing to ensure that the emic and the etic standpoints were better reflected.
Comment 9 : It is not clear enough why physical activity was not noticed — was it because of the visuals, the familiarity of the message, or the unrecognizable language? It remains unanalysed.
Response 9 : We interpret that the poster on physical activity provoked less reactions mainly because the message was more obvious for the study’s participants. While the message was more controversial in the poster on meat (related to the belief that the body needs meat proteins, especially in aging populations). We however did not specifically assess with patients why they reacted more to the meat poster and less to the physical activity one. It is therefore difficult to further analyse this discrepancy.
Comment 10: Consider using numbers instead of just P and GP. For example, P01, GP04... Maybe readers will connect the thoughts of the participants.
Response 10 : Thank you for this comment, we have added this precision in our manuscript. All citations are referred with specific labels, and these labels are reported in Table 1 which describes the study participants.
Comment 11 : Chapter 3.3 needs to be elaborated in more detail. Try to connect it with earlier chapters.
Response 11 : Thank you for this comment, we have revised this section and added a transitional sentence to improve its connection with the preceding content.
Page 13, paragraph 5, lines 506-512: Following the concerns about legitimacy, time constraints, and the risk of moralizing or politicizing climate-related messages in clinical settings, some participants who supported a co-benefits approach offered suggestions on how to better tailor the communication strategies. Several participants emphasized the importance of promoting preventive messages about health/environmental co-benefits, while avoiding guilt-inducing or moralizing “activist” tones.
Comment 12 : There is insufficient insight into the difference by socio-demographic groups. Answers are not analysed: who makes which suggestions? Do younger respondents suggest digital messages and older respondents suggest informational flyers? Do women suggest more emotional messages? That would enrich the conclusions about the creation of messages according to target groups.
Response 12 Thank you for your comment. We recognise the importance of analysing responses by socio-demographic groups in order to understand how social positions influence perceptions and practices. However, for this pilot study with a limited sample size, we were unable to aim for a great variability in the sampling of participants. The sampling was elaborated at the medical practice level and followed a convenience sampling procedure (see Table 1 for description of medical practices). We were not able to ensure sufficient diversity in the participants’ profiles, because we observed patients who were present in the waiting room during the observation days. Information on sociodemographic profiles were collected during individual interviews, but not systematically, and this is a limitation of this study that we acknowledge in the limitations section. Thus, the limited availability of sociodemographic information at the individual level did not enable accounting for social dimensions in the analysis. We have better specified the sampling procedure in the methods section and added this aspect in the limitations section > see response and changes to Comment 4.
Comment 13: The financial dimension is mentioned but sparingly. The question of who is going to pay for the additional part of the consultation is extremely important — but not elaborated enough.
Response 13 : Thank you for this comment, it is indeed an important factor explaining the reluctance of discussing co-benefits in the clinical setting. In line with a comment from reviewer 1, we have added the following element in the limitations section (change in red):
Page 19, paragraph 5, lines 809-813:
The study also has limitations. The findings must be contextualized within the specific setting of general medicine practice in private clinics in Western Switzerland and may not be generalizable to other settings. As stated, general medicine practices in Switzerland are private, and although health insurance is compulsory to ensure universal coverage, patients face varying premiums and co-payments levels. The limited acceptance of co-benefits discussions during consultations – partly paid out-of-pockets – may be specific to this health financing scheme and may not apply to other settings.
Comment 14: Consider an explanation of the broader framework in which primary health care is implemented. I just read in the discussion that it is private in Switzerland. This way of organizing presents a special challenge for the implementation of such campaigns.
Response 14 : Again, this is a very good point, it has also been raised by another reviewer. We have described the specific political, economic and health system setting of Switzerland, that is to be taken into account for the external validity of the findings. The following elements are now described throughout the paper (new elements in red):
Page 16, paragraph 2, lines 630-637: In Switzerland, health discourse is deeply embedded in a rationale of individual responsibility, where individuals are expected to manage their own health through personal choices and behaviours - how they eat, move, and consume throughout their life [25].
Page 16, paragraph 4, lines 652-656: We situate this position in the specific context of the Swiss healthcare system, where GPs operate as liberal professionals in private practices, often detached from broader public health frameworks.
Section 4.1, page 19, paragraph 2, lines 807-813: The findings must be contextualized within the specific setting of general medicine practice in private clinics in Western Switzerland and may not be generalizable to other settings. As stated, general medicine practices in Switzerland are private, and although health insurance is compulsory to ensure universal coverage, patients face varying premiums and co-payments levels. The limited acceptance of co-benefits discussions during consultations – partly paid out-of-pockets – may be specific to this health financing scheme and may not apply to other settings.
Comment 15: State the limitations of this research.
Response 15 : We agree. We have now made a specific Strengths and limitations section. The following elements are stated as limitations:
Section 4.1. Page 19-20, lines 798-820:
4.1 Strengths and Limitations
A key strength of this study is that it gave voice to both patients and GPs on the topic of discussing co-benefits in medical settings. It enabled revealing the difficulty GPs face in positioning themselves in a dualistic perception of the private vs. public, and the scientific vs. political – dimensions that had not emerged through survey data [8]. It also included patients from diverse walks of life, as they were recruited and invited to speak in-vivo in their GP’s waiting room. Such diversity is often not captured in surveys, due to recruitment biases.
The study also has limitations. The findings must be contextualized within the specific setting of general medicine practice in private clinics in Western Switzerland and may not be generalizable to other settings. As stated, general medicine practices in Switzerland are private, and although health insurance is compulsory to ensure universal coverage, patients face varying premiums and co-payments levels. The limited acceptance of co-benefits discussions during consultations – partly paid out-of-pockets – may be specific to this health financing scheme and may not apply to other settings.
The sample of interviewed GPs and patients was limited in number, and the diversity of GPs perspectives may be constrained by selection bias. Participants' profiles were not collected systematically during interviews, limiting the exploration of how sociodemographic positions influenced viewpoints. Finally, participants’ awareness of the study's environmental focus may have influenced their responses, potentially introducing desirability bias. Individuals with sceptical views on climate change may be underrepresented.
Comment 16: The discussion should be better organized. The authors mention terms such as "ambivalence" and "political perception", but do not elaborate on them conceptually.
Response 16 : As stated in response 8, we have fully revised the results and discussion sections, to ensure a better organisation. Terms such as ambivalence or political perception have been illustrated more clearly.
Comment 17: There is a lack of deeper reflection on mechanisms of influence: Why would positive messages work more? Does it have anything to do with cognitive processing, motivation, identity? It is possible to refer here to theories of health communication.
Response 17 : Thank you for your comment. As this is a pilot study, we have not included an analysis referring to theories of health communication. This could be explored further in a future study. Although we are a multidisciplinary research team, we do not have an expertise in theories of health communication, and this brings a limitation for a deeper reflection on the mechanisms of influence indeed. We acknowledge this in our conclusion:
Page 20, paragraph 3, lines 836-844:
These insights point to the need for interdisciplinary approaches in clinical practice to develop and evaluate innovative tools that promote dialogue on the links between health and the environment. Further research should consider drawing on behavioural theory and exploring structural changes needed to support transitions toward increased co-benefits. This includes considering how environmental issues can be effectively integrated into the concerns of public health and the medical system.
Comment 18: Failures of the intervention are not explicitly discussed. The intervention obviously did not achieve the main goal — recognition of co-benefit messages and their spontaneous introduction into the conversation. The authors relativize it. Suggestion: openly admit design failures and include reflection on visual design, ambiguity of messages, and unpreparedness of GPs for these topics.
Response 18 : We agree that we can better acknowledge that the gap between the attitudes of GPs (reported in André et al’s study) and the practice tested in this study can be better acknowledged. We however think that findings such as the debate on meat consumption/reduction is a useful finding that highlights the need for GPs to prepare and position themselves for a medical conversation. Revising the discussion section, we have made cleared statements about what worked and what did not work, specifying what can be linked to the specific context of Switzerland, and what can be explained by messaging that was perceived as negative. We have revised the discussion and made these elements of visual design, ambiguity of message and unpreparedness of GPs more prominent.
See Discussion and particularly : Page17-18, last paragraph, lines 714-721 : This study underscores the necessity of formulating materials that are aligned, as much as possible, with the diverse opinions surrounding climate change. The content and visual design of the posters provoked negative responses among those less familiar with, or less convinced by, climate-related issues. It therefore failed to engage them in a discussion about co-benefits. In this sense, this study revealed shortcomings in both visual design and messaging, which could have been mitigated through a stronger co-production strategy. However, this does not diminish the finding that GPs were limited in their ability to address such topics with patients.
Comment 19: Above all, the text should be changed to make it easier to read. Enter chapters. Visualization. Also stick to the good old approach of one paragraph dealing with one topic. Where possible, shorten the text and avoid excessive elaboration. The authors have quite enough material that they should present.
Response 19 : We have reviewed and edited the English language (using Copilot AI for support). We have added Table 1 to better visualise the study participants and settings. We have reviewed the results section, adding sub-titles to facilitate navigation. We have fully revised the discussion section to ensure a better understanding of the emic and etic standpoints, and we have paid attention to paragraph editing.
Reviewer 3 Report
Comments and Suggestions for Authors
The manuscript under review is the result of highly relevant and current research from studies across diverse fields, which allows the study to transcend aspects of the health sciences.
By addressing the relationship between health and the environment from the perspective of physicians and patients in Primary Heath care, the authors introduce the environmental theme in a transdisciplinary manner and with pedagogical application.
Advances in scientific knowledge are supported, among other reasons, by the comparison of results obtained from two groups: physicians and patients.
To broaden the understanding of the results, it is important that the authors describe the patients' level of education. Since the aim is to measure participants' knowledge of the subject under study, we know that education directly impacts the results.
The list of bibliographic references appears to comply with the journal's guidelines. In conclusion, I affirm that the study is relevant and merits publication in a scientific publication.
Author Response
Comment 1 : To broaden the understanding of the results, it is important that the authors describe the patients' level of education. Since the aim is to measure participants' knowledge of the subject under study, we know that education directly impacts the results.
Response 1 : Thank you for reviewing our manuscrit and for your pertinent comment. Due to the specific nature of the medical waiting room context, it was not possible to collect the personal information of participants during the observations. Some information was collected during the interviews, but not systematically. We acknowledge this limitation of our study and have included it in our paper. We have also added a summary table of participants' profiles in section 2.3.
Page 20, paragraph 1, lines 816-817: Participants' profiles were not collected systematically during interviews, limiting the exploration of how sociodemographic positions influenced viewpoints.
Page 5, line 173:
Table 1 : Participants’ profiles
|
Patients’ profiles’ information |
General Practitioners’ profiles |
|||||
|
Observations |
Interviews |
|||||
|
Type of medical practice |
Number of patients |
Information on patients’ profiles |
Information on patients’ profiles |
|||
|
A. Urban medical practice (canton de Neuchâtel) |
8 women and 3 men |
The observed patients tended to be older. |
P03 |
A woman in her forties from Eastern Africa, works as a housecleaner. |
GP01 |
A 50-year-old woman practicing as a family physician. |
|
P04 |
A woman in her sixties with a university degree in Literature. |
|||||
|
P05 |
A man in his thirties with a GP partner. |
|||||
|
B. Urban medical practice (canton de Vaud) |
5 women and 5 men |
The observed patients were diverse in age and multicultural. |
|
|
GP03 |
A 47-year-old woman practicing as a family physician with additional qualifications in acupuncture and Chinese pharmacotherapy. |
|
GP04 |
A 57-year-old man practicing as a family physician with additional expertise in infectious diseases, tropical medicine, and travel medicine. |
|||||
|
C. Urban medical practice (canton de Vaud) |
7 women and 1 man |
The observed patients were predominantly female, with an age range of 40 to 80 years, and primarily sought acupuncture treatment. |
|
|
|
|
|
D. Urban medical practice (canton de Vaud) |
2 women and 3 men |
The clientele was, with an age range of approximately 40 to 80 years. |
P01 |
A retired nurse and humanitarian aid worker from France. |
GP02 |
A 51-year-old man practicing as a family physician. |
|
E. Medical practice in rural area (canton de Vaud) |
4 women and 3 men |
The clientele was comprised of individuals between the ages of 40 and 85. |
P02 |
A man in his sixties from Italy, working in communications. |
GP05 |
A 54-year-old woman practicing as a family physician. |